# Bta-miR-206 and a Novel lncRNA-lncA2B1 Promote Myogenesis of Skeletal Muscle Satellite Cells via Common Binding Protein HNRNPA2B1

**DOI:** 10.3390/cells12071028

**Published:** 2023-03-27

**Authors:** Junxing Zhang, Hui Sheng, Linlin Zhang, Xin Li, Yiwen Guo, Yimin Wang, Hong Guo, Xiangbin Ding

**Affiliations:** 1Tianjin Key Laboratory of Agricultural Animal Breeding and Healthy Husbandry, Tianjin 300384, China; 2College of Animal Science and Veterinary Medicine, Tianjin Agricultural University, Tianjin 300384, China

**Keywords:** bovine, MuSCs, miR-206, lncRNA, HNRNPA2B1, myogenesis

## Abstract

Skeletal muscle satellite cells (MuSCs) can proliferate, differentiate, and self-renew, and can also participate in muscle formation and muscle injury repair. Long noncoding RNAs (lncRNAs) can play an important role with the RNA binding protein and microRNAs (miRNAs) to regulate the myogenesis of bovine MuSCs, however, its molecular mechanism is still being explored. In this study, differentially expressed 301 lncRNAs were identified during the myogenic differentiation of cells based on an in vitro model of induced differentiation of bovine MuSCs using RNA sequencing (RNA-seq). Based on the ability of miR-206 to regulate myogenic cell differentiation, a new kind of lncRNA-lncA2B1 without protein-coding ability was found, which is expressed in the nucleus and cytoplasm. Subsequently, lncA2B1 inhibited cell proliferation by downregulating the expression of the proliferation marker Pax7 and promoted myogenic differentiation by upregulating the expression of the differentiation marker MyHC, whose regulatory function is closely related to miR-206. By RNA pulldown/LC-MS experiments, heterogeneous ribonucleoprotein A2/B1 (HNRNPA2B1), and DExH-Box Helicase 9 (DHX9) were identified as common binding proteins of lncA2B1 and miR-206. Overexpression of lncA2B1 and miR-206 significantly upregulated the expression level of HNRNPA2B1. Downregulation of HNRNPA2B1 expression significantly decreased the expression level of the differentiation marker MyHC, which indicates that miR-206 and lncA2B1 regulate myogenic differentiation of bovine MuSCs by acting on HNRNPA2B1. This study screened and identified a novel lncRNA-lncA2B1, which functions with miR-206 to regulate myogenesis via the common binding proteins HNRNPA2B1. The results of this study provide a new way to explore the molecular mechanisms by which lncRNAs and miRNAs regulate muscle growth and development.

## 1. Introduction

Skeletal muscle cells originate as lumpy masses that divide after the formation of the embryonic mesoderm and ectoderm [1]. During the developmental stages of skeletal muscle, myogenesis determines the fate of muscle cells and eventual muscle formation [2,3]. Muscle satellite cells are activated and differentiated into myogenic cells. After several rounds of proliferation, the adult myoblasts exit the cell cycle and form myocytes. Myocytes form multinucleated myotubes by fusion and form myofibers [4]. The process of myogenesis is regulated by a variety of transcription factors, such as myogenic factor five (Myf5), myogenic differentiation (MyoD), myogenin (MyoG), and myogenic regulatory factor four (MRF4), as well as a growing number of noncoding RNAs (ncRNAs) and RNA-binding proteins (RBPs) [5]. The ncRNAs include microRNAs (miRNAs), long noncoding RNAs (lncRNAs), and circular RNAs (circRNAs), which are extensively involved in processes, such as the activation, proliferation, differentiation, and self-renewal of satellite cells [6,7,8,9]. However, the network of ncRNAs regulating the proliferation and differentiation of bovine skeletal muscle satellite cells and muscle growth, and development remains largely unknown. Further exploration is needed.

Many muscle-specific miRNAs have been identified and characterized, including miR-1, miR-206, and miR-133 families, regulating muscle production due to regulation by myogenic transcription factors [9,10,11]. miRNAs differentially expressed during myogenic differentiation of bovine skeletal muscle satellite cells (MuSCs) were screened and identified using miRNAs microarray technology, which included miR-206 [12]. miR-206 downregulated Pax7 by targeting its three prime untranslated regions (3′ UTR) and induced myoblast differentiation [11,12,13]. Understanding of the complex skeletal muscle developmental networks is broadened by discovering the molecule mechanisms of miRNA [14].

lncRNAs are more than 200 nt in length and have limited or no protein-coding capacity [15]. lncRNAs can participate in muscle growth and development by recruiting various signaling factors, altering chromosome conformation, and mediating or regulating the expression of myogenic regulatory factors [16]. In 1999, Lanz R B first identified lncRNA SRA [17], which acts as a molecular scaffold for p68/P72 and MyoD [18] and can mediate the transcriptional activation of target genes by MyoD and promote muscle differentiation. lncRNAs regulate cell growth, development, and differentiation by interacting with miRNAs. As the precursor of miR-675, lncRNA H19 can inhibit the activation and proliferation of hematopoietic stem cells [19]. As the competitive endogenous RNA (ceRNA) of miR-125b, lncMD can reduce the inhibitory effect of miR-125b on the IGF2 gene and promote the differentiation of bovine skeletal muscle [20]. lncRNAsYam-1, Malat1, linc-RAM, and linc-MD1, all involved in myogenic differentiation, were reported one after another [21].

Heterogeneous ribonucleoprotein A2/B1 (HNRNPA2B1) is a conservatively expressed RBP in mammals. It plays an important role in many biological processes, including the maturation, transport, and metabolism of messenger RNA (mRNA) and regulating the function of noncoding RNA [22]. After interfering with the expression level of HNRNPA2B1, cyclin D1, cyclin E, and CDC25A inhibit the proliferation of hESC and induce cell cycle arrest in the G0/G1 phase before differentiation. Knockdown of HNRNPA2B1 reduced the number of alkaline phosphatase-positive cells in hESCs and decreased the expression levels of pluripotency-associated transcription factors OCT4, NANOG, and SOX2, which indicates that HNRNPA2B1 is necessary for hESC self-renewal and pluripotency [23]. Single-cell RNA sequencing of regenerated skeletal muscle showed that neuromuscular diseases were related to HNRNPA2B1, and HNRNPA2B1 controlled the fate transformation of myogenic cells during terminal differentiation [24].

The present study aimed to identify novel lncRNAs associated with miR-206 regulation in bovine MuSCs and investigate their regulatory network and function during myogenesis in bovine MuSCs to provide a basis for elucidating the regulatory mechanism of bovine myogenesis.

## 2. Materials and Methods

### 2.1. Cell Isolation, Identification, and Culture

All tissue samples in this study were collected from a cattle farm in Fengzhen, Inner Mongolia, China, and met the national feeding standard NT/T815-2004. According to previous studies, we isolated, identified, and cultured primary bovine skeletal muscle satellite cells [12,25,26]. In brief, experimental cattle were euthanized by intravenous administration of an overdose of barbiturates. Bovine skeletal muscle satellite cells were isolated by digestion of hind limb muscle tissue from 5–6 month-old bovine fetuses with type II collagenase (Gibco, Grand Island, NY, USA) and trypsin (Gibco, Grand Island, NY, USA). The isolated cells were cultured in the growth medium containing 80% Dulbecco’s modified Eagle’s medium (DMEM, Gibco, Grand Island, NY, USA) and 20% fetal bovine serum (FBS, Gibco, Grand Island, NY, USA) until the cell density reached about 70%, then the differentiation medium containing 98% DMEM and 2% horse serum (HS, Gibco, Grand Island, NY, USA) was used to induce differentiation. Based on a previously constructed model of in vitro myogenic differentiation of bovine MuSCs [12,25], cells cultured for 24 h in a growth medium (GM) and cells cultured for 72 h in a differentiation medium (DM) for RNA sequencing (RNA-seq). All protocols of animal experiments were approved by the Institute of Animal Science of Tianjin Agricultural University, Tianjin, China, and were carried out in accordance with strict ethical policy (permit number TJAU20180312; 22 March 2018).

### 2.2. Total RNA Isolation and Reverse Transcription-qPCR (RT-qPCR)

Total RNA was extracted from bovine MUSCs and tissues using Trizol reagent (Invitrogen, Waltham, MA, USA). The integrity and concentration of the RNA samples were determined by agarose gel electrophoresis and NanoDrop 2000c (Thermo-scientific, Waltham, MA, USA). For miRNA detection, DNase-treated total RNA was reverse-transcribed to complementary DNA (cDNA) using the All-in-One^TM^ miRNA qRT-PCR assay kit (GeneCopoeia, Rockville, MD, USA). To detect lncRNA and mRNAs, the first strand cDNA was prepared using Prime SCRIPT II 1st Strand cDNA synthesis kit (Takara, Dalian, China). The All-in-One™ qRT-PCR Mix (Genocopoeia, Guangzhou, China) reaction system was used for qRT-PCR, together with forward primers, reverse primers, and cDNA. The 5.8s ribosomal RNA (rRNA) was used as an internal reference for miRNAs, and glyceraldehyde 3-phosphate dehydrogenase (GAPDH) was used as an internal reference for lncRNAs and mRNAs. The relative expression was detected by the 2-ΔΔCt method. All the primers used in this study are listed in Appendix A.

### 2.3. lncRNA Library Construction and Primary Analysis

Approximately 5 ug of total RNA was used to consume ribosomal RNA according to the manufacturer’s instructions of the Epicentre Ribo-Zero Gold Kit (Illumina, San Diego, CA, USA). Following purification, the RNA was fragmented into small pieces at high temperatures using divalent cations. The cleaved RNA fragments were then reverse transcribed according to the protocol of the RNA-Seq sample preparation kit (Illumina, San Diego, CA, USA) to create the final cDNA library. We performed paired-end sequencing (125 nt) on an Illumina Hiseq2500 sequencer at LC Biotech (Hangzhou, China) following the vendor’s recommended protocol.

Raw data (raw reads) of FASTQ format were first processed through in-house perl scripts. First, clean data (clean reads) were obtained by removing reads containing an adapter, reads containing ploy-N, and low-quality reads from raw data using cutadapt [27]. Then, the sequence quality was verified using FastQC (http://www.bioinformatics.bab-raham.ac.uk/projects/fastqc/) (accessed on 8 April 2018). All the downstream analyses were based on clean data with high quality.

A Tophat package [28] was used to obtain clean paired-end reads by mapping data to the UCSC (http://genome.ucsc.edu/) (accessed on 13 April 2018) cow reference genome (Bos_taurus_UMD_3.1.1). Cufflink [29] was used to de novo assemble the transcriptome; Cuffmerge was used to comerge all transcripts of samples to generate unique transcripts. Based on the transcript analysis, those similar or identical to the known transcripts were screened and classified by Cuffcompare comparison. Among the remaining transcripts, the coding potential prediction was performed using a coding potential calculator (CPC) [30] and a coding-noncoding index (CNCI) [31] to finally obtain the lncRNA candidates. The transcripts with a length ≥ 200 bp, exon ≥ 2, CPC score  <  − 1, and a CNCI score  <  0 were defined as novel lncRNAs.

Transcript abundance estimation and differentially expressed testing of the aligned read files were processed by Cufflinks. The unit of measurement is fragment per kilobase of exon per million fragments mapped (FPKM). The downloaded UCSC GTF file was passed to Cuffdiff along with the original alignment (SAM) file produced by Tophat. Cuffdiff reestimates the abundance of the transcripts listed in the GTF file using alignments from the SAM file and concurrently tests for different expressions. The |log2(flod change| ≥ 1 and *p* value < 0.05 was set as the threshold for significantly differential expression. The information on basic sequencing data is listed in Appendix A.

### 2.4. Generation of Constructs and Small Interfering RNA (siRNA) Synthesis

To construct the in vitro translation vector, lncA2B1 was subcloned into the EcoR I and Xho I sites of the pET-30a vector, and the pET-CNR encoding 27 kD protein was used as a positive control [32]. lncA2B1 was subcloned into Xho I and Xba I sites of pc DNA3.1 (+) vector to obtain recombinant overexpression vector pcDNA3.1-lncA2B1. The sequences of all primers used for gene cloning are listed in Appendix A. All siRNAs used for gene knockdown were designed and synthesized by RiboBio (Guangzhou, China), and the sequences are listed in Appendix A.

### 2.5. In Vitro Translation

The constructed recombinant expression vectors (pET-lncA2B1 and pET-CNR) were transferred into the BL21 receptor cells. Under the conditions of 37 °C at 200 rpm, the single clone was inoculated in a liquid culture medium and cultured until the optical density (600 nm) changed to 0.6~0.8. IPTG with the final concentration of 0.5 mM was then added and cultured for 2~4 h. Samples were centrifuged at 12,000 rpm for 1 min. The supernatant was discarded, and the precipitate was resuspended in 50–100 μL of 10 mM Tris-HCl (pH 8.0), followed by sodium dodecyl sulphate-polyacrylamide gel electrophoresis (SDS-PAGE).

### 2.6. Cell Transfection

Inhibitors, mimics, siRNAs, or plasmids were transferred into bovine skeletal MuSCs with Lipofetamine 3000 (Invitrogen, Waltham, MA, USA) according to the manufacturer’s instructions and the cell density was cultured to 60–70%. Final concentrations of 100 nM, 50 nM, 100 nM, and 2.0 µg/mL were used for inhibitors, mimics, siRNA, and plasmid agents respectively. GM phase and DM phase cells were collected separately for testing.

### 2.7. The 5-Ethynyl-2′-Deoxyuridine (EdU) Assay

After transfection for 24 h, bovine skeletal MuSCs were cultured using EdU reagent (RiboBio, Guangzhou, China) at 37 °C for 2 h. Cells were fixed with 4% paraformaldehyde (Solarbio Science and Technology Ltd., Beijing, China) for 30 min and neutralized with a 2 mg/mL glycine solution. Apollo^®^ staining solution containing EdU was added and incubated in the dark for 30 min at room temperature to label the DNA at the synthetic stage, followed by staining of the nuclei with 4′,6-diamidino-2-phenylindole (DAPI). Three images were randomly obtained using an inverted fluorescent microscope (Leica, Wetzlar, Germany) at a magnification of 400×, and the number of EdU-positive cells was counted.

### 2.8. Western Blot Analysis

The total protein of bovine MuSCs was collected using a radioimmunoprecipitation assay (RIPA) lysate containing protease inhibitor and phosphatase inhibitor (Takara, Dalian, China). The concentration of total cellular protein was determined using a BCA kit (CWbiotech Ltd., Beijing, China), analyzed by 10% SDS-PAGE, and transferred to PVDF membranes (Millipore, Burlington, MA, USA). The PVDF film was sealed with 5% BSA for 1 h, incubated overnight with the first antibody at 4 °C, incubated with the second antibody at room temperature for 1 h, and detected with ECL liquid exposure substrate (Solarbio Science and Technology Ltd., Beijing, China). The gray values of each protein band were calculated using ImageJ software. The antibody information is listed in Appendix A.

### 2.9. Nuclear-Cytoplasmic Fractionation

Cytoplasmic and nuclear extracts were isolated from myoblasts and myotubes using NE-PER Nuclear and Cytoplasmic Extraction Reagents (Thermo Scientific, Waltham, MA, USA), and all nuclear and cytoplasmic RNA/protein was extracted using TRIzol reagent/RIPA lysate [33]. Cells were collected; ice-cold CER I containing RNase inhibitor/PMSF was added; ice-cold CER II was added after 10 min of incubation on ice. After centrifugation at 16,000× *g* for 5 min, the cytoplasmic RNA/protein in the supernatant was extracted by Trizol/RIPA. The precipitate was washed 3 times with PBS and the nuclear RNA/protein in the precipitate was extracted by Trizol/RIPA.

### 2.10. Immunofluorescence Staining

Bovine MuSCs were fixed in 4% paraformaldehyde for 30 min, permeabilized in PBS containing 0.1% Triton X-100 for 20 min, closed with 5% BSA for 30 min, and incubated with anti-MyHC primary antibody at 4 °C overnight. Then, the second antibody labeled with FITC was added and incubated in the dark at 37 °C for 1 h. The nucleus was stained with DAPI and photographed with an inverted fluorescence microscope.

### 2.11. In Vitro Transcription

The recombinant plasmid pcDNA3.1-lncA2B1 was linearised and a template with a T7 promoter was prepared for in vitro transcription. The sense and antisense strands of biotin-labeled lncA2B1 were transcribed in vitro by T7 RNA polymerase. The DNA template in the system was eliminated by DNaseI, followed by RNA purification and electrophoresis detection. All primers used were listed in Appendix A.

### 2.12. Dual-Luciferase^®^ Reporter Assay System

lncA2B1 was cloned into Xho I and not I sites of the psicheck-2 vector (Promega, Medison, WI, USA), and then transferred into competent cells. Clones were screened with a solid LB medium containing 0.1% ampicillin, and the plasmid concentrations and masses were measured and stored at −20 °C. The sequence of primers is listed in Appendix A. Luciferase reporter vector, corresponding mimic, and lncRNA overexpression vector were cotransfected into 293T cells. After 48 h of culture, the cells were collected, and the samples were detected by a Dual-Luciferase^®^ Reporter Assay System (Promega, Medison, WI, USA).

### 2.13. lncRNA Pulldown/Liquid Chromatography–Mass Spectrometry (LC-MS)

The Pierce™ Magentic RNA-protein Pulldown Kit (Thermo-scientific, Waltham, MA, USA) was used to capture potential binding proteins to lncA2B1. The sense and antisense strands labeled with biotin were treated with an RNA structural buffer (10 mM Tris-HCl pH 7.0, 0.1 M KCl, and 10 mM MgCl_2_), and then incubated with RNA capture buffer, magnetic beads, and 1U/mLRNase inhibitors at 4 °C for 2 h. DM phase cells were collected using an IP lysis buffer containing protease inhibitors, and incubated with RNA-magnetic bead complexes for 2 h at 4 °C. The RNA-protein bead complexes were washed, mixed with a protein loading buffer, and incubated at 100 °C for 10 min.

The supernatant was subjected to SDS-PAGE, and the supernatant was identified by LC-MS. The mass spectrometry data was collected using the Triple TOF 5600 + LC/MS system (SCIEX, Redwood City, CA, USA). The raw analysis of the mass spectrometry data was partly done by Genecreate Biological Engineering Co., Ltd. (Wuhan, China). The original MS/MS files were submitted to ProteinPilot (https://sciex.com.cn/products/software/proteinpilot-software, version 4.5, SCIEX, Redwood City, CA, USA) (accessed on 05 February 2019) for data analysis. For protein identification, the Paragon algorithm in ProteinPilot was used to search the Bovine_uniprot database. The parameters were set as follows: the instrument is TripleTOF 5600, and the cysteine is modified with iodoacetamide; the biological modification is selected as the ID focus. For the identified protein results, select certain filtering criteria, and peptides with an unused score > 1.3 (credibility of more than 95%) are considered credible peptides, and proteins containing at least one unique peptide are retained. The information on basic protein data is listed in Appendix A.

### 2.14. miR-206 Pulldown/LC-MS

The negative control probes of 5 ’biotin-labeled miR-206 and mi-NC (Gene Pharma, Shanghai, China) were designed and synthesized. The probes were transfected with bovine MuSCs. The induced differentiated cells were collected, and the total protein was extracted. The magnetic beads were resuspended and centrifuged. The supernatant was removed, and the magnetic beads were washed with a washing buffer. The total protein (1 mg) was incubated with magnetic beads at 4 °C for 2 h, and the supernatant was collected after 3000 rpm centrifugation for 1 min. Finally, the supernatant was subjected to SDS-PAGE, and the supernatant was identified by LC-MS.

LC-MS analysis was performed on a nano Acquity UPLC system (Waters Corporation, Milford, MA, USA) connected to an LTQ Orbitrap XL mass spectrometer (Thermo Scientific, Bremen, Germany) equipped with an online nanoelectrospray ion source (Michrom Bioresources, USA). The mass spectra generated by the LTQ-XL instrument were processed using MaxQuant software (version 1.4.0.8, http://www.maxquant.org/) (accessed on 1 February 2019). Data were searched using the Andromeda search engine against the Bovine_uniprot database. The parameters for the database search were set as follows: (1) The minimum required peptide length was seven amino acids. (2) Trypsin cleavage specificity was applied with up to one missed cleavage allowed. (3) Variable modifications included methionine oxidation (M). (4) The mass tolerance for precursor and fragment ions was set to 20 ppm and 0.6 Da, respectively. (5) The false discovery rate (FDR) was set to 1% at both the peptide and protein levels. (6) Only proteins sequenced with at least two peptides were considered a reliable identification. The information on basic protein data is listed in Appendix A.

### 2.15. Statistical Analysis

All the results were expressed as the average value of ± standard error of the mean (SEM) in three independent experiments. Statistical analyses of differences between groups were performed using a two-tailed Student’s *t*-test or chi-square test, and *p* < 0.05 was considered statistically significant (* *p* < 0.05 and ** *p* < 0.01).

## 3. Results

### 3.1. Identification of Differentially Expressed lncRNAs during Myogenic Differentiation of Bovine MuSCs

To enrich and expand the understanding of the role of lncRNA in the myogenic differentiation of bovine MuSCs, cells cultured in a growth medium for 24 h (GM) and cells cultured in a differentiate medium for 72 h (DM) for RNA-Seq were collected. Compared with GM phase cells, 301 differentially expressed lncRNAs were identified in the DM phase cells (Figure 1a), of which 213 lncRNA expression was upregulated, and 88 lncRNA expression levels were downregulated (Figure 1b). Clustering analysis of these 301 lncRNAs showed that they were differentially expressed in the GM phase and DM (Figure 1c).

### 3.2. Screening and Identification of lncRNAs Associated with miR-206

A bovine MuSCs model with overexpression or knockout of miR-206 was constructed by transfecting mimics or inhibitors (Figure 1d). When miR-206 was overexpressed, the expression level of the proliferation marker Pax7 protein was significantly downregulated (*p* < 0.01), while the expression level of the differentiation marker MyHC protein was significantly upregulated (*p* < 0.01) (Figure 1e). When interfering with the expression of miR-206, the expression level of Pax7 protein was significantly upregulated (*p* < 0.01), while the expression level of MyHC protein was significantly downregulated (*p* < 0.01) (Figure 1f). To screen for lncRNAs associated with miR-206, an association analysis of miR-206 and 301 differentially expressed lncRNAs was performed by predicted target genes, and 22 of them were associated. A clustering analysis of these 22 lncRNAs was performed. Twelve lncRNAs were upregulated, and 10 lncRNAs were downregulated during the DM phase (Figure 1g). To verify the reliability of the sequencing data, the expression levels of candidate lncRNAs (|log2(flod change| ≥ 2) were examined in the retained sequencing samples using RT-qPCR. TCONS_00024351, TCONS_00070803, lncA2B1, and TCONS_00109062 were significantly differentially expressed in GM- and DM-phase cells, which were consistent with the trend of the sequencing results (Figure 1g,h). lncA2B1 (No. of bases < 3,000) was significantly upregulated in DM-phase cells (*p* < 0.01) (Figure 1h), and overexpression of miR-206 significantly upregulated the expression level of lncA2B1 (*p* < 0.01) (Figure 1i), and we selected lncA2B1 for followup studies.

### 3.3. lncA2B1 Is a Long Noncoding RNA

The genomic information of lncA2B1 was obtained through the online software Ensembl and NCBI. lncA2B1 was transcribed from chromosome two of the bovine genome and was composed of four exons. The four exon sequences of lncA2B1 overlap with the unidentified LOC100847374 gene and are named lncA2B1 (Figure 2a). The first exon overlapped with the ACTR3 gene, but lncA2B1 was transcribed in the opposite direction to ACTR3. The genes surrounding lncA2B1 included ACTR3, SLC35F5, and LOC112442697 (Figure 2a). The lncA2B1 sequence was verified authentic by RT-PCR across exons and sequencing of the amplified fragments (Figure 2b). This study predicted the protein-coding potential of lncA2B1 by CPC and CPC2. It had no coding potential (−0.92744), comparable to the known linc-MD1 (−0.893097), however, it had a 46 AA open reading frame ORF (Figure 2c) (Appendix A). The ORF was predicted to encode a <15 KD protein. The prokaryotic expression vector pET-lncA2B1 did not express the protein in vitro translation experiments, while the positive control pET-CNR encoded a 27 KD protein, which verifies that lncA2B1 does not have the protein-coding ability (Figure 2d). In this study, the expression level of lncA2B1 in the cytoplasm and nucleus of GM-phase and DM-phase bovine MuSCs were detected, and lncA2B1 was expressed in the cytoplasm and nucleus (Figure 2e).

### 3.4. Role of lncA2B1 in the Proliferation of Bovine MuSCs

The regulatory effects of lncA2B1 on cell myogenic proliferation were investigated by constructing bovine MuSCs models that overexpress and interfere with lncA2B1 (Figure 2f,g) (Appendix A). The mRNA and protein expression levels of Pax7 were significantly downregulated after the overexpression of lncA2B1 (*p* < 0.01) (Figure 2f). When interfering with the expression of lncA2B1, the mRNA and protein expression levels of Pax7 were significantly upregulated (*p* < 0.01) (Figure 2g). The proliferation assay of 5-ethynyl-2′-deoxyuridine (EdU) cells showed that the number of EdU-positive cells and EdU labeling index were decreased in bovine MuSCs overexpressing lncA2B1 (*p* < 0.05) (Figure 2h). Interference with lncA2B1 expression significantly increased the number of EdU-positive cells and EdU labeling index (*p* < 0.05) (Figure 2h).

### 3.5. Role of lncA2B1 in the Myogenic Differentiation of Bovine MuSCs

In the present study, the regulatory role of lncA2B1 on cellular myogenic differentiation was further investigated by constructing bovine MuSCs models overexpressing and interfering with lncA2B1 (Figure 3a,b). Overexpression of lncA2B1 significantly upregulated the mRNA expression of differentiation marker gene MyoG (*p* < 0.05) (Figure 3a), increased the protein expression of MyHC and MyoG (*p* < 0.05) (Figure 3c), and thickened the myotubes formed by cells (*p* < 0.01) (Figure 3d,e). In contrast, interference with lncA2B1 expression resulted in a significant downregulation of the mRNA expression levels of MyHC (*p* < 0.01) (Figure 3b), a significant reduction in the protein expression levels of MyHC and MyoG (*p* < 0.05) (Figure 3c), a reduction in the number of myotubes formed by the cells and a poorer trend of cell fusion (*p* < 0.01) (Figure 3d,e). Immunofluorescence staining showed that overexpression of lncA2B1 led to an increased proportion of green fluorescence specific for MyHC protein and increased the cell fusion index (*p* < 0.01). In contrast, interference with lncA2B1 decreased the proportion of green fluorescence specific for the MyHC protein and reduced the cell fusion index (*p* < 0.01) (Figure 3f,g).

### 3.6. lncA2B1 May Function Associated with miR-206

Through the association analysis by RNAhybrid and combination prediction by the online tool RegRNA 2.0 of miR-206 and lncA2B1, it has been found that there is a binding energy mfe = −25.4 kcal/mol between them (Figure 4a). Cotransfection of miR-206 mimic and psicheck2-lncA2B1 into 293T cells significantly reduced the relative activity of Renilla luciferase by a dual luciferase assay (*p* < 0.01) (Figure 4b). When 0.1 μg or 0.2 μg of pcDNA3.1-lncA2B1 was added, pcDNA3.1-lncA2B1 significantly restored luciferase activity in a dose-dependent manner (*p* < 0.05) (Figure 4b). The results of the double luciferase reporting system showed that there was a binding relationship between miR-206 and lncA2B1. Analysis of the regulation of expression between miR-206 and lncA2B1 by cotransfection experiments revealed that the expression levels of lncA2B1 and miR-206 were significantly increased (*p* < 0.01) in GM cells transfected with pcDNA3.1-lncA2B1 alone or cotransfected with pcDNA3.1-lncA2B1 and miR-206 mimic. Meanwhile, the Pax7 and HDAC4 protein expression levels were downregulated (Figure 4c). The Pax3, Pax7, and HDAC4 mRNA expression levels also were downregulated (Figure A1a). In GM cells transfected with si-lncA2B1 alone or cotransfected with si-lncA2B1 and miR-206 inhibitor, interference with miR-206 did not affect the expression level of lncA2B1, whereas interference with lncA2B1 reduced the expression level of miR-206 and upregulated the protein expression levels of Pax7 and HDAC4 (Figure 4d). Also, the Pax3, Pax7, and HDAC4 mRNA expression levels also were upregulated (Figure A1b). In GM cells transfected with si-lncA2B1 alone or cotransfected with si-lncA2B1 and miR-206 mimic (Figure A1c,d), and in cells transfected with pcDNA3.1-lncA2B1 alone or cotransfected with pcDNA3.1-lncA2B1 and miR-206 inhibitor (Figure A1e,f), suggesting that lncA2B1 may promote muscle formation in bovine MuSCs together with miR-206.

### 3.7. Acquisition of lncA2B1 Binding Protein

To further explore the specific mechanism between lncA2B1 and miR-206, the pulldown experiment of lncA2B1 was designed. The full-length DNA chain of lncA2B1 was amplified, and the RNA probe was obtained by in vitro transcription (Figure 4e). The silver staining results of the pulldown experiment showed a difference between the just and antisense groups (Figure 4e). Protein strip samples from both groups were examined by LC-MS and retrieved and analyzed using protein-pilot software. When the confidence level was 95%, 45, and 34 proteins were identified in the sense and antisense groups, respectively. According to the Venn analysis results, lncA2B1 specifically bound 20 proteins (Figure 4f).

### 3.8. Acquisition of miR-206 Binding Protein

The pulldown experiment of miR-206 was designed (Figure 4g). The miR-206 DNA double-stranded probe containing 5′- biotin and its negative control DNA double-stranded probe were transfected into bovine MuSCs, and the biotin-labeled probe was used to capture the proteins that could bind to miR-206. The bands between the miR-206 and the mi-NC groups differed significantly (Figure 4g). LC-MS analysis showed that 491 proteins were captured by the 5′Biotin-miR-206 probe, and 381 proteins were captured by the 5′Biotin-mi-NC probe, of which 172 proteins were specifically bound by miR-206 (Figure 4h).

### 3.9. lncA2B1 and miR-206 act on the Binding Proteins HNRNPA2B1

By Venn analysis of these specific binding proteins and protein blotting experiments, HNRNPA2B1 and DHX9, are common binding proteins for miR-206 and lncA2B1(Figure 4i,j). In this study, the specific binding protein HNRNPA2B1 was selected for indepth studies, and some of the results for DHX9 were added to the supporting information (Figure A2). The expression levels of HNRNPA2B1 in cells were examined by overexpressing and interfering with lncA2B1 and miR-206, respectively. When lncA2B1 was overexpressed, the mRNA (Figure 5a) and protein levels (Figure 5c) of HNRNPA2B1 were significantly increased (*p* < 0.05). Interference with lncA2B1 revealed no significant changes in the mRNA expression levels (Figure 5b) and protein expression levels (Figure 5c) of HNRNPA2B1. When miR-206 was overexpressed, both mRNA and the protein levels of HNRNPA2B1 were significantly upregulated (*p* < 0.05) (Figure 5d). After interfering with miR-206, the mRNA expression level of HNRNPA2B1 did not change (*p* > 0.05) (Figure 5e), and the protein expression level decreased significantly (*p* < 0.05) (Figure 5f).

### 3.10. HNRNPA2B1 Functions in the Myogenesis of Bovine MuSCs

To verify the effect of HNRNPA2B1 on myogenesis, si-HNRNPA2B1 was transfected into bovine MuSCs (Figure 5g). The mRNA level and protein expression level of Pax7 were increased, though the difference was not significant (*p* > 0.05). The proliferation assay of EdU cells showed no change in the number of EdU-positive cells and EdU labeling index in bovine MuSCs when HNRNPA2B1 interfered (Figure 5h). When the expression of HNRNPA2B1 was downregulated, the expression level of miR-206 was also decreased (*p* < 0.01). The expression level of lncA2B1 did not change significantly (Figure 5i). We found the m6A recognizable RNA motif of HNRNA2B1 in the lncA2B1 sequence (Figure 5j). Interference with HNRNA2B1 expression resulted in a poorer trend of cell fusion (Figure 6a) and a significant downregulation of the mRNA (*p* < 0.01) and protein expression (*p* < 0.05) levels of MyHC (Figure 6b). The expression level of the HNRNPA2B1 protein at the DM phases was increased (*p* < 0.05), especially in the nucleus (*p* < 0.01) (Figure 6c). These results suggest that miR-206 and lncA2B1 regulate the myogenic differentiation of MuSCs by interacting with the common binding protein HNRNPA2B1 (Figure 7). Preliminary results from this study showed that lncA2B1 regulated miR-206 expression and affected target gene Pax3/Pax7/HDAC4 expression to inhibit cell proliferation. lncA2B1 binds more to the HNRNPA2B1 protein to regulate the expression of the differentiation marker MyHC, promoting myogenic differentiation.

## 4. Discussion

Development and differentiation of skeletal muscle satellite cells are essential in developmental biology research. As myogenic stem cells, MuSCs play an important role in repairing and regenerating skeletal muscle injury [34]. lncRNAs are becoming the gatekeepers of MuSC proliferation and differentiation [26]. lncRNA MUNC can promote the function of MyoD in skeletal muscle formation [35]. In mouse myogenic cells, lnc-31 is associated with myogenic cell proliferation and inhibits cell differentiation during myogenesis [36]. In this study, 301 differentially expressed lncRNAs were identified during the myogenic differentiation of bovine MuSCs using RNA-seq (Figure 1a,b). According to a previous study, miR-206 plays a vital role in the myogenesis of bovine skeletal muscle mesenchymal stem cells (Figure 1e,f) [12]. Therefore, the relationship between miR-206 and 301 lncRNAs was analyzed, and a novel lncRNA-lncA2B1 was identified through a model of in vitro-induced differentiation in bovine MuSCs.

Different organelle-localized lncRNAs have different regulatory mechanisms [37,38,39]. In the present study, the expression level of lncA2B1 in the DM phase is 3.9 times higher than in the GM phase (Figure 1h), and lncA2B1 was mainly located in the cytoplasm during the GM phase and in the nucleus during the DM phase (Figure 2e); it is suggested that lncA2B1 may play a role through transmembrane movement or significant upregulation in the nucleus of the DM phase. lncRNA can regulate the translocation of HNRNPs from nuclear to cytoplasm [40,41]. The binding protein expression level of HNRNPA2B1 at the DM phases was increased, especially in the nucleus. Therefore, the combination of lncA2B1 and HNRNPA2B1 regulates myoblast differentiation, which may be related to the change in the subcellular localization of lncA2B1.

When exploring the function of genes, the biological function of candidate genes is usually studied by using the expression level of overexpressed or knocked-out candidate genes [21]. Jiang et al. [42] analyzed the regulatory mechanisms of pyruvate dehydrogenase B (PDHB) on muscle differentiation using primary skeletal muscle cell models in humans and mice, with knockdown or overexpression assays. In this study, MuSCs models were constructed that overexpressed and interfered with lncA2B1; lncA2B1 downregulated the mRNA and protein expression levels of Pax3/Pax7/HDAC4, the number of EdU-positive cells were decreased (Figure 2f–h and Figure A1a,b). lncA2B1 significantly upregulated the mRNA expression of differentiation marker gene MyoG (Figure 3a), increased the protein expression of MyHC and MyoG (Figure 3c), and thickened the myotubes formed by cells (Figure 3d,e). The above results showed that lncA2B1 had a negative regulatory effect on the proliferation and a positive regulatory effect on the myogenic differentiation of bovine MuSCs. Further exploration of this study will require analysis through other myogenic cell models or mouse models of denervation.

Some studies had shown that lncRNAs may act as sponges of myoblast proliferation inducers miRNAs such as miR-206, miR-208, and miR-133 (binding energy mfe < −25.0 kcal) [43]. In this study, a binding energy mfe = −25.4 kcal between lncA2B1 and miR-206 was found (Figure 4a), suggesting that lncA2B1 is able to bind miR-206 and we verified their binding by a dual luciferase reporter assay (Figure 4b). The miRNAs microarray technique was used to screen and identify the differentially expressed miRNAs during the myogenic differentiation of bovine MuSCs, miR-206 was significantly upregulated in myotubes [44]. miR-206 positively regulate bovine MuSCs myogenic differentiation via Pax7 and HDAC4 downregulation [12]. Analysis of the regulation of expression between miR-206 and lncA2B1 by cotransfection experiments revealed that lncA2B1 might work together with miR-206 to promote muscle formation in bovine MuSCs by coinfluencing the expression levels of Pax3, Pax7, and HDAC4 (Figure 4c,d and Figure A1). lncA2B1 and miR-206 are not common ceRNA regulatory mechanisms. lncRNAs can modulate the stability and nuclear turnover of specific mRNAs via RBPs and miRNAs [45]. This mechanism sheds light on our study. Therefore, the experiments of lncA2B1 pulldown and miR-206 pulldown (Figure 4e–h) were designed. Two common binding proteins of miR-206 and lncA2B1, DHX9 and HNRNPA2B1, were identified (Figure 4i). Preliminary results from this study showed that lncA2B1 and miR-206 are closely related to regulatory function, and lncA2B1 regulated miR-206 expression and affected target gene Pax3/Pax7/HDAC4 expression to inhibit cell proliferation.

HNRNPA2B1 is known to form complexes with lncRNAs and is emerging as an important mediator of lncRNA-induced transcriptional repression [46,47]. lncRNA RP11 can form an RP11-HNRNPA2B1-mRNA complex with HNRNPA2B1 and downregulate the mRNA stability of Siah1 and Fbxo45 in CRC cells [48]. The lnc-HC interacts with HNRNPA2B1 to form an lnc-HC-HNRNPA2B1 complex, which targets the mRNA of Cyp7a1 and Abca1 to reduce protein expression [49]. The results of this study show that lncA2B1 can pull down the HNRNPA2B1 protein (Figure 4i,j), and lncA2B1 overexpression upregulated the expression level of HNRNPA2B1 (Figure 5a,c). Therefore, lncA2B1 may exert regulatory effects by binding to HNRNPA2B1 and forming an RNA-protein complex.

HNRNPA2B1 can bind to m6A and RGAC motifs in nuclear transcripts labeled by the RNA methyltransferase METTL3 [50]. The m6A/METTL3 methylation mark allows for the effective recognition of pri-miRNAs by DGCR8 and their subsequent processing to pre- and mature miRNAs. The expression level of miR-206 was decreased in mouse embryonic stem cells with METTL3 as the CRISPR target [51]. HNRNPA2B1 can act as an adaptor that recruits the microprocessor complex to a subset of precursor miRNAs, facilitating their processing into mature miRNAs [50,52,53]. When the expression of HNRNPA2B1 was knocked down, the expression level of miR-206 decreased (Figure 5i), and the recognition site of m6A on lncA2B1 was found (Figure 5j). The results suggest that HNRNPA2B1 recognizes lncA2B1 and miR-206 through the m6A motif. More recently identified roles for RBPs in myogenesis include maintenance of MuSCs quiescence [54], MuSC activation and expansion [55,56], and myogenic differentiation [57,58]. HNRNPA2B1 regulates cell proliferation [46] and differentiation [23,59]. In this study, when the expression of HNRNPA2B1 was knocked down, the mRNA and protein expression levels of MyHC were significantly downregulated (Figure 6a,b), which was consistent with previous studies [24]. Preliminary results from this study showed that lncA2B1 binds more to the HNRNPA2B1 protein to regulate the expression of the differentiation marker MyHC, promoting myogenic differentiation.

DHX9 participates in the process of lncRNA biogenesis [60]. lncRNA-SH3PXD2A-AS1 interacted with DHX9 to enhance FOXM1 expression, promote tumor cell proliferation, and accelerate cell cycle progression [61]; lncRNA-KIMAT1 binds to and stabilizes DHX9, this effect is attributable to proteasomal degradation [62]. lncA2B1 was able to pull down (Figure 4i,j) and affect the expression level of (Figure A2) DHX9 protein, suggesting that lncA2B1 might bind to and stabilize DHX9 protein. DHX9 can bind to the RISC complex and participate in the silencing of target genes by miRNA [63,64]. When DHX9 was knocked down, the miR-206 expression level was reduced, and the Pax7 protein expression level was improved (Figure A2g–i). We speculated that DHX9 binds to miR-206, and activates RISC silencing of Pax7, thereby regulating the myogenesis differentiation of bovine MuSCs. The detailed molecular mechanism is not currently understood in depth and still requires further validation.

## 5. Conclusions

A novel lncRNA-lncA2B1 was screened, which is related to the function of miR-206. HNRNPA2B1 was identified as a common binding protein of miR-206 and lncA2B1 by RNA pulldown/LC-MS. It elucidates the role of miR-206 and lncA2B1 in promoting the myogenesis of bovine MuSCs through the common binding protein HNRNPA2B1. This study provides new insights into the molecular mechanisms by which lncRNAs and miRNAs regulate muscle growth and development.

## Figures and Tables

**Figure 1 cells-12-01028-f001:**
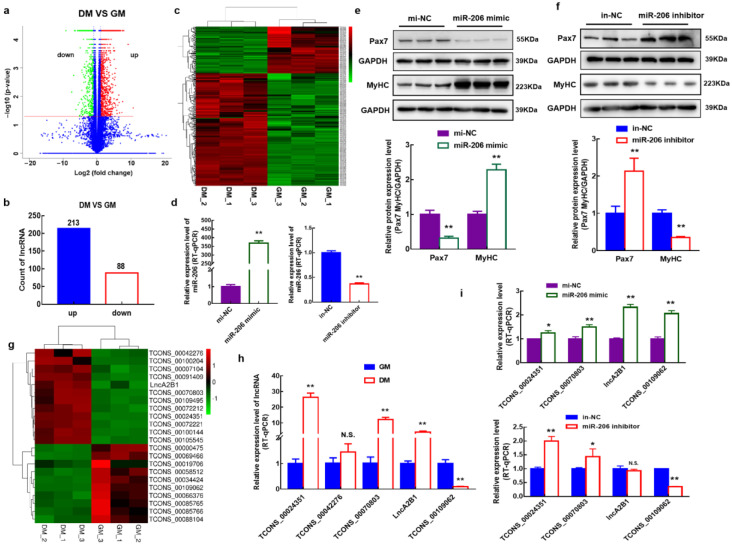
Screening and identification of myogenesis-associated lncRNA. (**a**) Identification of differentially expressed lncRNAs in GM phase and DM phase bovine MuSCs. GM: cells cultured in growth medium for 24 h, DM: cells cultured in differentiation medium for 72 h. (**b**) The up- and downregulated expression distribution of differentially expressed lncRNA. (**c**) Cluster analysis was performed on 301 differentially expressed lncRNA. (**d**) A bovine MuSCs model which overexpressed or interfered with miR-206 was successfully established. (**e**,**f**) The protein expression levels of Pax7 and MyHC were detected by western blot. (**g**) Heat map of the expression levels of 22 lncRNA associated with miR-206 before and after differentiation. (**h**) Differential expression of candidate lncRNAs in GM-phase and DM-phase bovine MuSCs detected by RT-qPCR. (**i**) Overexpression or interference with miR-206 affects the expression levels of lncRNA. *, *p* < 0.05; **, *p* < 0.01; N.S., *p* > 0.05.

**Figure 2 cells-12-01028-f002:**
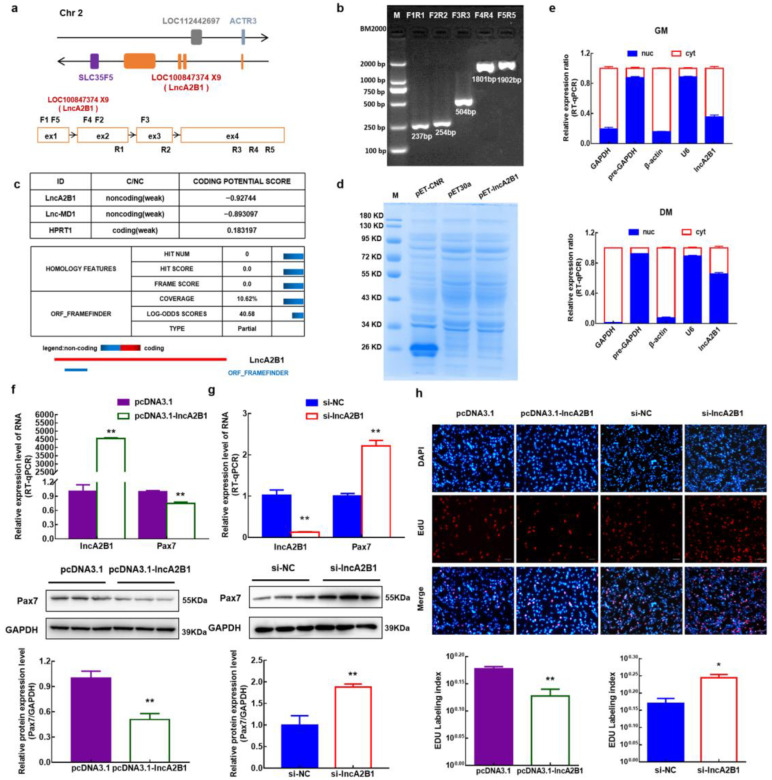
lncA2B1 is a noncoding long RNA that affects the proliferation of bovine MuSCs. (**a**) Genomic information for lncA2B1. (**b**) The base sequence of lncRNA was verified by RT-PCR detection and monoclonal sequencing. (**c**) The CPC program was used to predict the coding potential of genes. Both lncA2B1 and lnc-MD1 are predicted to be noncoding RNA, while HPRT1 (hypoxanthine phosphor ribosyl transferase 1) has the ability to encode proteins. (**d**) In vitro translation assays using the transcription factor CNR as positive control showed no protein product from lncA2B1. (**e**) The nucleoplasmic separation of GM-phase and DM-phase bovine MuSCs was carried out and subcellular localization was carried out by RT-qPCR. GAPDH or β-actin was used as cytoplasmic control and pre-GAPDH or U6 as nuclear control. (**f**,**g**) The effect of overexpression or downregulation of lncA2B1 on the expression of mRNA and protein of proliferation marker Pax7. (**h**) EdU staining was used to detect the effect of lncA2B1 on the proliferation of bovine MuSCs (×200, scale 100 μm). *, *p* < 0.05; **, *p* < 0.01, N.S., *p* > 0.05.

**Figure 3 cells-12-01028-f003:**
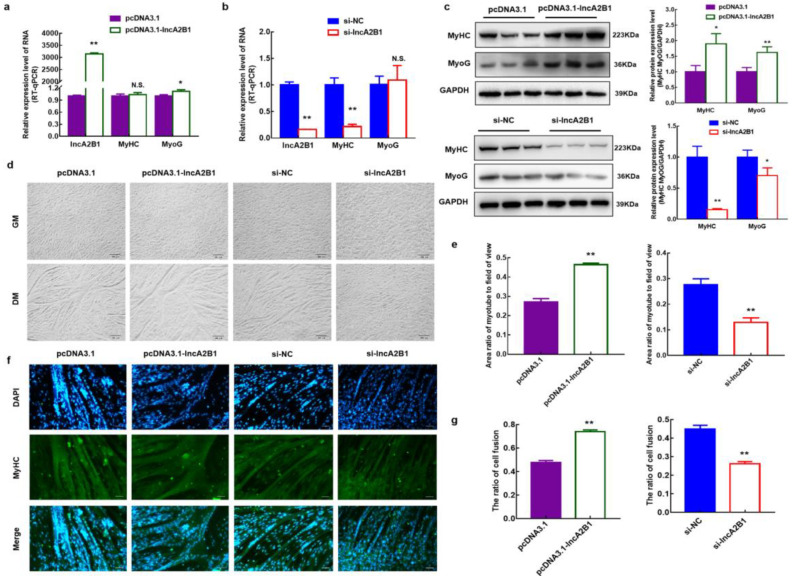
The lncA2B1 functions in the myogenic differentiation. (**a**,**b**) The cell differentiation models of overexpressing and interfering with LncA2B1 were established, and the mRNA expression levels of the differentiation marker genes MyHC and MyoG were detected by RT-qPCR. (**c**) The protein levels of MyHC and MyoG were detected by Western blot. (**d**,**e**) The Image J software was used to measure the area of myotube and field of view. The area ratio of the myotube to the field of view was performed to quantificationally evaluate myotube formation (×100, scar bar 200 μm). (**f**) The expression levels of MyHC in bovine MuSCs overexpressing and interfering with lncA2B1 were detected by immunofluorescence staining and the proportion of cells fused was measured using Image J software (×200, scar bar 100 μm). *, *p* < 0.05; **, *p* < 0.01; N.S., *p* > 0.05.

**Figure 4 cells-12-01028-f004:**
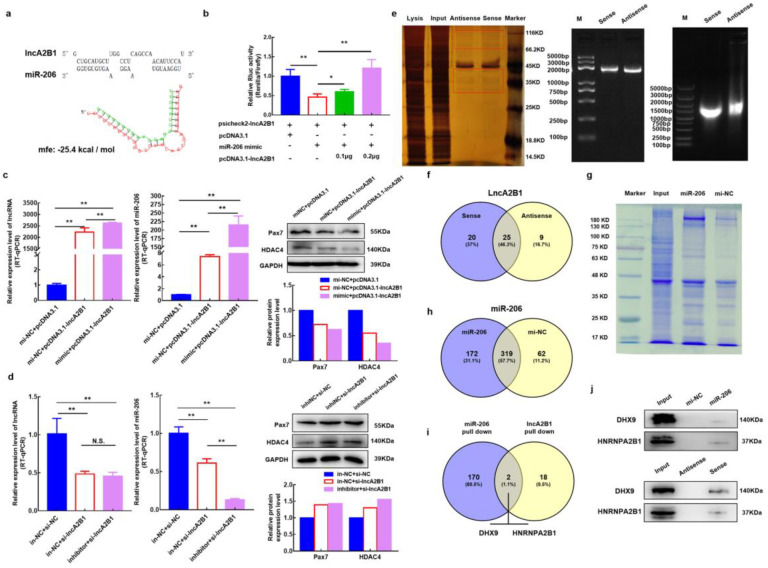
lncA2B1 may function associated with miR-206 and the acquisition of lncA2B1 and miR-206 cobinding proteins. (**a**) The targeted binding site of lncA2B1 and miR-206 is predicted by RNAhybrid and RegRNA 2.0. (**b**) Dual-luciferase assay validates lncA2B1 binding to miR-206. (**c**) Bovine MuSCs were transfected with pcDNA3.1-lncA2B1 and miR-206 mimic alone or together, and the RNA expression levels of lncA2B1 and miR-206, and the protein expression levels of Pax7 and HDAC4 were examined. (**d**) Bovine MuSCs were transfected with si-lncA2B1 and miR-206 inhibitors alone or together, and the RNA expression levels of lncA2B1 and miR-206 as well as the protein expression levels of Pax7 and HDAC4 were examined. (**e**) Interacting proteins with lncA2B1 were obtained by RNA pulldown. Lysis: the total protein of the lysate; Input: the protein remaining after the probe of the experimental group is enriched. (**f**) Venn analysis of antisense and sense-bound proteins, 20 proteins specifically bound by sense. (**g**) SDS-PAGE gels of proteins interacting with miR-206 were obtained by RNA pulldown. (**h**) Venn analysis of miR-206 and mi-NC bound proteins, 172 proteins specifically bound by miR-206. (**i**) DHX9 and HNRNPA2B1 were identified as common proteins that bind to miR-206 and lncA2B1. (**j**) Validation of the reliability of DHX9 and HNRNPA2B1 in RNA pulldown mass spectrometry using western blot. *, *p* < 0.05; **, *p* < 0.01; N.S., *p* > 0.05.

**Figure 5 cells-12-01028-f005:**
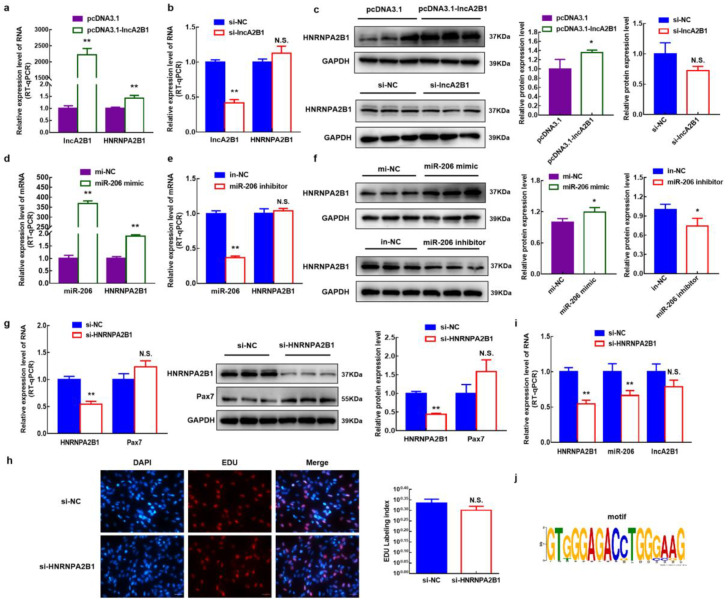
The lncA2B1 and miR-206 regulated myogenesis via binding protein HNRNPA2B1. (**a**–**c**) Overexpressed lncA2B1 significantly upregulated the mRNA and protein expression level of HNRNPA2B1; Interference with lncA2B1 did not affect the mRNA and protein expression level of HNRNPA2B1. (**d**–**f**) Overexpressed miR-206 significantly upregulated the mRNA and protein expression level of HNRNPA2B1; Interference with miR-206 did not affect the mRNA expression level of HNRNPA2B1, significantly downregulated the protein expression. (**g**) Interference with HNRNPA2B1 did not affect the mRNA and protein expression levels of Pax7 in GM. (**h**) EdU staining was used to detect the effect of si-HNRNPA2B1 on the proliferation of bovine MuSCs (×400, scale 50 μm). (**i**) Expression of miR-206 was significantly downregulated, and lncA2B1 did not change significantly when HNRNPA2B1 interfered. (**j**) The RGAC site of lncA2B1 by motif analysis, which is a known recognition motif for HNRNPA2B1. *, *p* < 0.05; **, *p* < 0.01, N.S., *p* > 0.05.

**Figure 6 cells-12-01028-f006:**
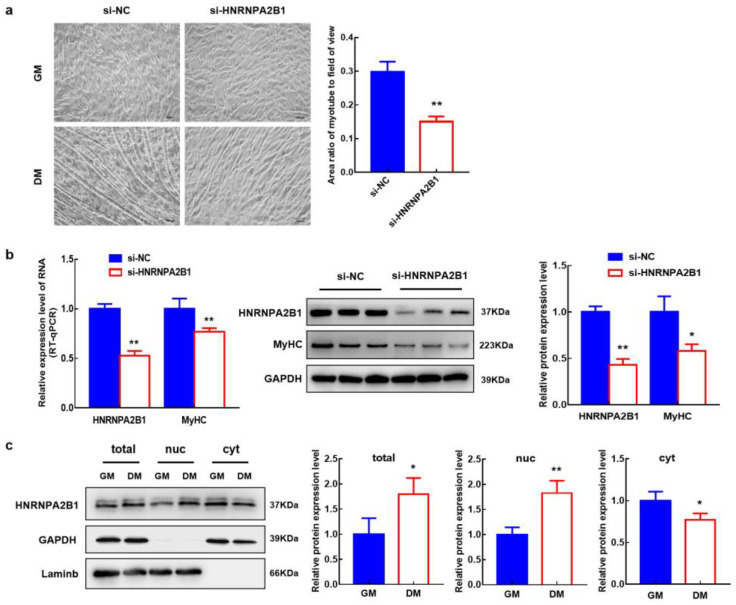
HNRNPA2B1 functions in the myogenesis of bovine MuSCs. (**a**) The Image J software was used to measure the area of myotube and field of view. Then, the area ratio of the myotube to the field of view was performed to quantificationally evaluate myotube formation (×200, scar bar 100 μm). (**b**) Interference with HNRNPA2B1 significantly inhibited the mRNA and protein expression levels of MyHC in DM. (**c**) The nucleoplasmic separation of GM- and DM-phase bovine MuSCs was carried out, and the protein HNRNPA2B1 was subcellularly localized by western blot. GAPDH is used as cytoplasmic control and Laminb as nuclear control. *, *p* < 0.05; **, *p* < 0.01.

**Figure 7 cells-12-01028-f007:**
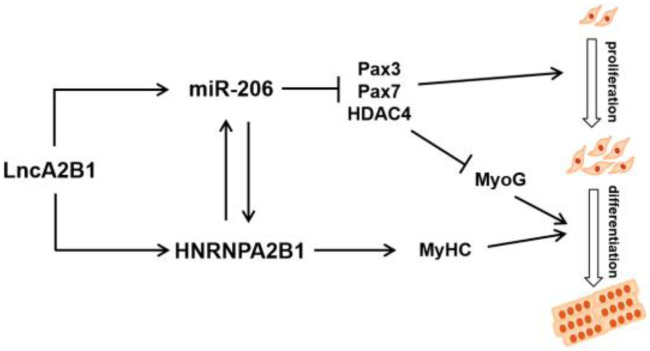
The molecular mechanisms of miR-206 and lncA2B1 function in myogenesis.

## Data Availability

All of the data generated or analyzed during this study are included in this published article and its Appendix A.

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
