# Peer review of "Bta-miR-206 and a Novel lncRNA-lncA2B1 Promote Myogenesis of Skeletal Muscle Satellite Cells via Common Binding Protein HNRNPA2B1"

_cells, 2023, doi:10.3390/cells12071028_

Round 1
Reviewer 1 Report
This manuscript authored by Zhang J. and colleagues present an interesting study of non-coding RNAs and RNA-binding proteins in bovine skeletal muscle satellite cells. In this study, miR-206, lncA2B1 and HNRNPA2B1 are found to work together to regulate muscle growth and development. The current manuscript is well prepared, and the results are consistent and supportive to the conclusion. Some concerns should be addressed to improve the current study.
1. One single model, the MuSCs culturing, was used to test the hypothesis in the entire study. It seems that the scientific findings have a very limited biological significance.
2. It is unknown if miR-206 was detected and analyzed in the RNA-seq. What was the change if there was any?
3.Though Pax7 and EdU staining are appropriate approach studying cell proliferation, cell counting or cell cycle analysis may provide more direct evidence for changes in cell growth.
4. No evidence is provided to support direct interaction or functional dependence between miR-206 and LncA2B1.
5. The results to unveil the molecular mechanism underlying the roles of miR-206, lncA2B1 and HNRNPA2B1 in cell proliferation and differentiation are very limited.
6. Minor concern. The name of the newly identified lncRNA should appear in the title.
Author Response
Dear Reviewers,
Thank you for your letter. Thanks very much for taking your time to review this manuscript. We thank the reviewers for the time and effort that they have put into reviewing the previous version of the manuscript. Your suggestion shave enabled us to improve our work. Based on the instructions provided in your letter, we uploaded the file of there vised manuscript. Accordingly, we have uploaded a copy of the original manuscript with all the changes highlighted by using the track changes mode in MS Word.
Appended to this letter is our point-by-point response to the comments raised by the reviewers. The comments are reproduced and our responses are given directly afterward in a different color (blue).
Thanks again!
Reviewer #1
This manuscript authored by Zhang J. and colleagues present an interesting study of non-coding RNAs and RNA-binding proteins in bovine skeletal muscle satellite cells. In this study, miR-206, lncA2B1 and HNRNPA2B1 are found to work together to regulate muscle growth and development. The current manuscript is well prepared, and the results are consistent and supportive to the conclusion. Some concerns should be addressed to improve the current study.
Response: Thank you for your affirmation of the contents, results and conclusions of the manuscript. We have revised the shortcomings of the manuscript and responded accordingly point-by-point. Thank you for reviewing it again. I look forward to receiving your positive reply, and thank you again for your hard work in reviewing the manuscript.
1. One single model, the MuSCs culturing, was used to test the hypothesis in the entire study. It seems that the scientific findings have a very limited biological significance.
Response: Thank you very much for your suggestion. Our research group has been doing a lot of work based on bovine skeletal muscle satellite cells [1-7], and has found differentially expressed genes in the muscle tissues of fetal cows at different months, as well as in the muscle tissues of fetal cows and adult cows, and verified them through cell models. Therefore, in this study, we focus on the regulation mechanism in the process of myogenesis by directly using primary bovine skeletal muscle satellite cells. In this study, gene regulation can be found more directly and effectively through the difference of two periods of cell model.
Using other primary cells or in vivo experiments to further verify the results of this study was a challenge for us. Although our research group has a research foundation for the in vivo experimental mouse muscle denervation model, it will take a long time and energy. We haven't made any progress in this field, which needs to be considered in the follow-up study. We will also compare with the results of this study and share them with readers.
We have added the limitations of this study in the discussion section of the manuscript. Further exploration of this study will require analyzed through other myogenic cell models or mouse models of denervation.
[1] Sheng H, Guo Y, Zhang L, Zhang J, Miao M, Tan H, Hu D, Li X, Ding X, Li G, Guo H. Proteomic Studies on the Mechanism of Myostatin Regulating Cattle Skeletal Muscle Development. Front Genet. 2021 Nov 16;12:752129. doi: 10.3389/fgene.2021.752129.
[2] Zhang X, Chen M, Liu X, Zhang L, Ding X, Guo Y, Li X, Guo H. A novel lncRNA, lnc403, involved in bovine skeletal muscle myogenesis by mediating KRAS/Myf6. Gene. 2020 Aug 15;751:144706. doi: 10.1016/j.gene.2020.144706.
[3] Chen M, Li X, Zhang X, Li Y, Zhang J, Liu M, Zhang L, Ding X, Liu X, Guo H. A novel long non-coding RNA, lncKBTBD10, involved in bovine skeletal muscle myogenesis. In Vitro Cell Dev Biol Anim. 2019 Jan;55(1):25-35. doi: 10.1007/s11626-018-0306-y.
[4] Dai Y, Zhang WR, Wang YM, Liu XF, Li X, Ding XB, Guo H. MicroRNA-128 regulates the proliferation and differentiation of bovine skeletal muscle satellite cells by repressing Sp1. Mol Cell Biochem. 2016 Mar;414(1-2):37-46. doi: 10.1007/s11010-016-2656-7.
[5] Dai Y, Wang YM, Zhang WR, Liu XF, Li X, Ding XB, Guo H. The role of microRNA-1 and microRNA-206 in the proliferation and differentiation of bovine skeletal muscle satellite cells. In Vitro Cell Dev Biol Anim. 2016 Jan;52(1):27-34. doi: 10.1007/s11626-015-9953-4.
[6] Zhang WR, Zhang HN, Wang YM, Dai Y, Liu XF, Li X, Ding XB, Guo H. miR-143 regulates proliferation and differentiation of bovine skeletal muscle satellite cells by targeting IGFBP5. In Vitro Cell Dev Biol Anim. 2017 Mar;53(3):265-271. doi: 10.1007/s11626-016-0109-y.
[7] Jin CF, Li Y, Ding XB, Li X, Zhang LL, Liu XF, Guo H. lnc133b, a novel, long non-coding RNA, regulates bovine skeletal muscle satellite cell proliferation and differentiation by mediating miR-133b. Gene. 2017 Sep 30;630:35-43. doi: 10.1016/j.gene.2017.07.066.
2. It is unknown if miR-206 was detected and analyzed in the RNA-seq. What was the change if there was any?
Response: Thank you for your suggestion. miR-206 is significantly up-regulated in differentiated myotubes compare to bovine skeletal muscle satellite cells. In fact, it has been described in a previous study by our group, see [1-2]. We have added an explanation of this section to the discussion, thanks again.
[1] Wang, Y.M.; Ding, X.B.; Dai, Y.; Liu, X.F.; Guo, H.; Zhang, Y. Identification and bioinformatics analysis of miRNAs involved in bovine skeletal muscle satellite cell myogenic differentiation. Mol Cell Biochem 2015, 404, 113-122, doi:10.1007/s11010-015-2371-9.
[2] Dai, Y., Wang, Y.M., Zhang, W.R., Liu, X.F., Li, X., Ding, X.B., et al. (2016). The role of microRNA-1 and microRNA-206 in the proliferation and differentiation of bovine skeletal muscle satellite cells. In Vitro Cell Dev Biol Anim 52(1), 27-34. doi: 10.1007/s11626-015-9953-4.
3.Though Pax7 and EdU staining are appropriate approach studying cell proliferation, cell counting or cell cycle analysis may provide more direct evidence for changes in cell growth.
Response: Thank you very much for your suggestion. Your suggestion is of great significance for improving the level of the manuscript.
Cell counting or cell cycle analysis was indeed not designed in the previous trials, and we have great difficulty in supplementing the trials. It will take a long time to purchase the determination reagents, primary cell resuscitation, passage, transfection test and determination. At present, schools and laboratories are in the holiday stage. It is too late for us to complete this revision.
5-ethynyl-2’-deoxyuridine (EdU) is an alkyne-tagged thymidine analogue that incorporates into newly synthesized DNA during replication [1]. As a result of the copper-catalysed azide-alkyne cycloaddition reaction, i.e. “Click chemistry” with a fluorescent azide, EdU has been used in fluorescence imaging to follow DNA synthesis of proliferating cells [2-3]. This technique demonstrates the great advantage of not relying on an antibody to label DNA, omitting the need for the rigorous step of DNA denaturation [1]. This study analyzed the cell proliferation process and mainly analyzed the myogenic differentiation process. In some mechanism studies supplemented by cell proliferation experiments, some studies only used proliferation marker genes and EDU test or BrdU test [4].
I hope you understand. If the analysis of cell proliferation without cell counting or cell cycle analysis is not feasible, please email us and we will try to complete the supplement. Thank you again for your hard work.
[1] Salic, A.; Mitchison, T.J. A chemical method for fast and sensitive detection of DNA synthesis in vivo. Proc Natl Acad Sci U S A 2008, 105, 2415-2420, doi:10.1073/pnas.0712168105.
[2] Ishizuka, T.; Liu, H.S.; Ito, K.; Xu, Y. Fluorescence imaging of chromosomal DNA using click chemistry. Sci Rep 2016, 6, 33217, doi:10.1038/srep33217.
[3] Basile, D.P.; Friedrich, J.L.; Spahic, J.; Knipe, N.; Mang, H.; Leonard, E.C.; Changizi-Ashtiyani, S.; Bacallao, R.L.; Molitoris, B.A.; Sutton, T.A. Impaired endothelial proliferation and mesenchymal transition contribute to vascular rarefaction following acute kidney injury. Am J Physiol Renal Physiol 2011, 300, F721-733, doi:10.1152/ajprenal.00546.2010.
[4] Mir, B.A.; Albrecht, E.; Ali, A.; Hansson, O.; Maak, S. MicroRNA-100 Reduced Fetal Bovine Muscle Satellite Cell Myogenesis and Augmented Intramuscular Lipid Deposition by Modulating IGF1R. Cells 2022, 11,doi:10.3390/cells11030451.
4. No evidence is provided to support direct interaction or functional dependence between miR-206 and LncA2B1.
Response: Thank you very much for your suggestion. The design idea of this study was to try to find miRNAs and lncRNAs with correlation by microarray of miRNAs from cells before and after induction differentiation of bovine skeletal muscle satellite cells and RNA-seq (including mRNA and lncRNA). miR-206 is a small RNA that was found to be highly expressed during the period of cell differentiation in the group's preliminary study. We identified a number of novel lncRNAs that are differentially expressed during cell differentiation. lncA2B1 was identified through correlation analysis of their predicted target genes, which is highly expressed during cell differentiation and has a beneficial regulatory role in the process of myogenesis. We attempted to find a relationship between them by predicting and verifying their previous direct binding and regulatory relationships, and to find their co-regulatory proteins through RNA pull-down experiments. Obviously, we obtained a predicted mfe = -25.4 kcal between miR-206 and LncA2B1. It has been shown that lncRNAs may act as sponges of myoblast proliferation inducers microRNAs (miRNAs) such as miR-206, miR-208, and miR-133 (binding energy mfe < -25.0 kcal) [1]. It is suggested that lncA2B1 is able to bind miR-206 and we verified their binding by a dual luciferase reporter assay.
Although not a common ceRNA regulatory mechanism between miR-206 and lncA2B1, lncA2B1 was able to influence the expression of miR-206 as well as the target gene Pax7/Pax3/HDAC4, as revealed by the regulatory mechanism assay. Not only that, but we also identified their common binding proteins, HNRNPA2B1 and DHX9.
We thought that perhaps these tests were also still not sufficient to support an interaction between miR-206 and lncA2B1, and therefore a direct interaction or functional dependence between miR-206 and lncA2B1 was not reflected in the manuscript. miR-206 and lncA2B1 are closely related in regulatory function, which is a more accurate description.
In the discussion part, we supplement the discussion about the function between miR-206 and lncA2B1. Thank you again for your hard work.
[1] García-Pérez, I.; Molsosa-Solanas, A.; Perelló-Amorós, M.; Sarropoulou, E.; Blasco, J.; Gutiérrez, J.; Garcia de la serrana, D. The Emerging Role of Long Non-Coding RNAs in Development and Function of Gilthead Sea Bream (Sparus aurata) Fast Skeletal Muscle. Cells 2022, 11, 428.
5. The results to unveil the molecular mechanism underlying the roles of miR-206, lncA2B1 and HNRNPA2B1 in cell proliferation and differentiation are very limited.
Response: Thank you for your suggestions. Your suggestions have given us some enlightenment.
The present study aimed to identify novel lncRNAs associated with miR-206 regulation in bovine MuSCs and investigate their regulatory network and function during myogenesis in bovine MuSCs to provide a basis for elucidating the regulatory mechanism of bovine myogenesis.
First, this study found and identified a new lncRNA-lncA2B1 that can regulate the myogenesis process of bovine skeletal muscle satellite cells. In addition, the binding protein of miR-206 and lncA2B1 was found through RNA-pull down test, and a relatively reliable molecular mechanism was obtained through preliminary research and analysis. We think the content of this study is relatively rich and the results are reliable.
miR-206 and lncA2B1 regulate the myogenic differentiation of MuSCs by interacting with common binding protein HNRNPA2B1 (Figure 7). Preliminary results from this study showed that lncA2B1 regulated miR-206 expression and affected target gene Pax3/Pax7/HDAC4 expression to inhibit cell proliferation. lncA2B1 binds more to the HNRNPA2B1 protein to regulate the expression of the differentiation marker MyHC, promoting myogenic differentiation. This study provides new insights into the molecular mechanisms by which lncRNAs and miRNAs regulate muscle growth and development.
We have added some limitations of this study in the discussion section of the manuscript.
6. Minor concern. The name of the newly identified lncRNA should appear in the title.
Response: We thank the reviewer for raising these important points. We have added the name lncRNA-lncA2B1 to the title.
Reviewer 2 Report
The work by Zhang et al. with the title "Bta-miR-206 and a novel LncRNA promote myogenesis of skeletal muscle satellite cells via the common binding protein HNRNPA2B1", presents an exciting approach to understanding the role of lncRNAs in bovine myogenesis.
The study's conceptualization is exciting, a huge amount of experiments were performed, and the findings might be precious, although several issues must be addressed to validate the scientific soundness of the present work.
Material and methods overall are lacking information for example:
How it was confirmed the types of cell that were isolated was correct?
Where are the RNA-seq library preparation and bioinformatics methods used?
Was performed mass spectrometry? It was mentioned in the results, but it was not stated in the methods...
the nuclear and cytoplasmatic RNA enriched isolation, why 16000g was used, isn't that too much? Usually, the nuclear enriched part must be washed after the first centrifugation, that was done? More controls of nuclear and cytoplasmatic enriched RNAs must be shown.
Results.
Overall it is hard to follow while reading one must assume something was done making the read unpleasant, as a result of bad style writing and lack of essential details. The reader can have the impression that the results come from nowhere, and methods and reasoning of results are merged with results, making the reading slow and hard.
The discussion is too short and should be more broadly explained.
Overall Zhang et al. show a work with exciting results, that can be suitable for publication once the scientific soundness is proven.
Author Response
Dear Reviewers,
Thank you for your letter. Thanks very much for taking your time to review this manuscript. We thank the reviewers for the time and effort that they have put into reviewing the previous version of the manuscript. Your suggestion shave enabled us to improve our work. Based on the instructions provided in your letter, we uploaded the file of there vised manuscript. Accordingly, we have uploaded a copy of the original manuscript with all the changes highlighted by using the track changes mode in MS Word.
Appended to this letter is our point-by-point response to the comments raised by the reviewers. The comments are reproduced and our responses are given directly afterward in a different color (blue).
Thanks again!
Reviewer #2
The work by Zhang et al. with the title "Bta-miR-206 and a novel LncRNA promote myogenesis of skeletal muscle satellite cells via the common binding protein HNRNPA2B1", presents an exciting approach to understanding the role of lncRNAs in bovine myogenesis.
The study's conceptualization is exciting, a huge amount of experiments were performed, and the findings might be precious, although several issues must be addressed to validate the scientific soundness of the present work.
Response: Thank you for your hard work. We have revised the shortcomings of the manuscript and responded accordingly point-by-point. Thank you for reviewing it again. I look forward to receiving your positive reply, and thank you again.
Material and methods overall are lacking information for example:
1.How it was confirmed the types of cell that were isolated was correct?
Response: Thank you very much for your suggestion. We have included the first part of Materials and Methods for cell isolation and culture, adding some description of cell identification.
In the early stage, the research group did a lot of research on the methods of isolating and identifying skeletal muscle satellite cells, not only isolated and identified bovine skeletal muscle satellite cells, but also isolated and identified mouse and rabbit skeletal muscle satellite cells. The specific operation steps also condense a lot of previous research experience, and the method of isolation and identification of skeletal muscle satellite cells is published.
Please refer to the literature [1-3] for details of in vitro isolation and culture, induction of differentiation and identification.
[1] Wang, Y.M., Dai, Y., Liu, X.F., Liu, Z.W., Li, J.X., Guo, H., et al. (2014). Isolation, Identification and Induced Differentiation of Bovine Skeletal Muscle Satellite Cell. China Animal Husbandry & Veterinary Medicine 41(07), 142-147. doi: CNKI:SUN:GWXK.0.2014-07-033.
[2] Dai, Y., Wang, Y.M., Liu, X.F., et al. (2014). Isolated Culture and Identification of Mouse Skeletal Muscle Satellite Cell. Journal of Tianjin Agricultural University 21(01), 1-4.
[3] Dai, Y., Wang, Y.M., Liu, X.F., et al. (2014). Isolated Culture and Identification of rabbit Skeletal Muscle Satellite Cell. Heilongjiang Animal Science and Veterinary Medicine (21):176-179+238. doi:10.13881/j.cnki.hljxmsy.2014.21.058.
2. Where are the RNA-seq library preparation and bioinformatics methods used?
Response: Thank you very much for your suggestion. The RNA-seq library was prepared and sequenced by Hangzhou Lianchuan Biologicals, and the bioinformatics analysis was also done by our analysis department and platform. Is your suggestion that this section needs to be added to the material methods section? We have tried to collate this section from previous analysis reports and add it to the Materials and Methods section.
3.Was performed mass spectrometry? It was mentioned in the results, but it was not stated in the methods...
Response: Thank you very much for your suggestion. We have added a description of the mass spectrometry data analysis in section 2.13. The raw analysis of the mass spectrometry data was partly done by Genecreate Biological Engineering Co., Ltd. (Wuhan, China) and we performed a discrepancy analysis based on the mass spectrometry data results.
4. the nuclear and cytoplasmatic RNA enriched isolation, why 16000g was used, isn't that too much? Usually, the nuclear enriched part must be washed after the first centrifugation, that was done? More controls of nuclear and cytoplasmatic enriched RNAs must be shown.
Response: Thank you very much for your advice. Nuclear and cytoplasmic RNA extraction was conducted using NE-PER Nuclear and Cytoplasmic Extraction Reagents (Thermo Scientific, Waltham, MA) according to the manufacturer’s protocol. The instructions specify that this step involves centrifugation at maximum speed (~16, 000 x g) for 5 minutes and that the supernatant is then transferred for extraction. Insufficient centrifugation time or centrifugal force in this step would result in incomplete nuclear separation.
We supplemented more controls of nuclear and cytoplasmatic enriched RNAs, β-actin act as cytoplasmic and U6 act as nuclear controls, and the results were supplemented in Figure 2e. By the cytoplasmic-expressing GAPDH, nuclear-expressing pre-GAPDH and the supplemented controls in this study, all indicate that the nuclear and cytoplasmic separation of the cells was successful.
The nuclear enriched part was washed after the first centrifugation, using PBS 3 times, followed by the lysis of the nuclear RNA, which is described in the Materials and Methods supplement.
Results.
5. Overall it is hard to follow while reading one must assume something was done making the read unpleasant, as a result of bad style writing and lack of essential details. The reader can have the impression that the results come from nowhere, and methods and reasoning of results are merged with results, making the reading slow and hard.
Response: Thank you very much for your suggestion, and we have invited a native English-speaking editor to embellish the manuscript. At the same time, we have provided a Certification, which has been uploaded into the system. Thank you again for your hard work!
6 The discussion is too short and should be more broadly explained.
Response: Thank you very much for your suggestion. We have supplemented the discussion part of the manuscript.
Overall Zhang et al. show a work with exciting results, that can be suitable for publication once the scientific soundness is proven.
Round 2
Reviewer 1 Report
All the reviewer's questions have been answered. Good luck!
Author Response
Dear Reviewers,
Thank you for your letter. Thanks very much for taking your time to review this manuscript. Your suggestion shave enabled us to improve our work. Based on the instructions provided in your letter, we uploaded the vised manuscript. Accordingly, we have uploaded a copy of the original manuscript with all the changes highlighted by using the track changes mode in MS Word. The second revision was highlighted with a yellow background.
Appended to this letter detailing our changes and point-by-point response to the comments raised by the reviewers. The comments are reproduced and our responses are given directly afterward in a different color (blue).
Thanks again!
Reviewer #1:
All the reviewer's questions have been answered. Good luck!
Response: Thank you for your hard work in reviewing this manuscript. Good luck!
Reviewer 2 Report
Zhang et. al. answered most of the questions and the work has improved. However, part of science's soundness is due to reproducibility, bioinformatic analysis was a crucial part of the work and was poorly described, which inhibits the evaluation of how everything downstream was performed. RNA-seq and mass-spectrometry analysis must be well and clearly explained. What software and packages were used, which command flags were considered, what was the version of the genome used, and how new lncRNAs was predicted? Is the data going to be shared, where, and how?
After this issue is properly addressed I am keen to reevaluate the work.
Author Response
Dear Reviewers,
Thank you for your letter. Thanks very much for taking your time to review this manuscript. Your suggestion shave enabled us to improve our work. Based on the instructions provided in your letter, we uploaded the vised manuscript. Accordingly, we have uploaded a copy of the original manuscript with all the changes highlighted by using the track changes mode in MS Word. The second revision was highlighted with a yellow background.
Appended to this letter detailing our changes and point-by-point response to the comments raised by the reviewers. The comments are reproduced and our responses are given directly afterward in a different color (blue).
Thanks again!
Reviewer #2:
Zhang et. al. answered most of the questions and the work has improved. However, part of science's soundness is due to reproducibility, bioinformatic analysis was a crucial part of the work and was poorly described, which inhibits the evaluation of how everything downstream was performed. RNA-seq and mass-spectrometry analysis must be well and clearly explained. What software and packages were used, which command flags were considered, what was the version of the genome used, and how new lncRNAs was predicted? Is the data going to be shared, where, and how?
After this issue is properly addressed I am keen to reevaluate the work.
Response: Thank you very much for your suggestion, which is of great help to the improvement of the manuscript. We have revised the manuscript according to your suggestions.
The specific methods of RNA-seq and mass spectrometry analysis are described below. Due to the length of the following description, in view of the length of the manuscript, only the specific data analysis part has been added to the manuscript (the software and software packages used, the command flags considered and the genome version used have all been reflected in the revised manuscript).
The mass spectrometry systems after lncA2B1 and miR-206 pull down are inconsistent, so they are described respectively.
We have uploaded the basic data of each part to the supplementary materials. The description was added to the materials and methods. The raw data of RNA-seq has not been uploaded to the public database at present. Requests to raw data used should be directed to XB: xiangbinding@tjau.edu.cn.
RNA-seq analysis
Raw data (raw reads) of FASTQ format were firstly processed through in-houseperl scripts. Firstly, clean data (clean reads) were obtained by removing reads containing adapter, reads containing ploy-N and low-quality reads from raw data using Cutadapt [1]. Then sequence quality was verified using FastQC(http://www.bioinformatics.babraham.ac.uk/projects/fastqc/). All the downstream analyses were based on clean data with high quality.
Tophat package [2] was used to obtain clean paired-end reads by mapping data to the UCSC (http://genome.ucsc.edu/) cow reference genome (Bos_taurus_UMD_3.1.1). Cufflink [3] was used to de novo assemble the transcriptome; Cuffmerge was used to co-merge all transcripts of samples to generate unique transcripts. Based on the transcript analysis, those similar or identical to known transcripts were screened and classified by Cuffcompare comparison. Among the remaining transcripts, the coding potential prediction was performed using Coding Potential Calculator (CPC) [4] and Coding-Non-Coding Index (CNCI) [5] to finally obtain lncRNA candidates. The transcripts with length≥200 bp, exon≥2, CPC score < − 1 and CNCI score < 0 were defined as novel lncRNAs.
Transcript abundance estimation and differentially expressed testing the aligned read files were processed by Cufflinks. The unit of measurement is Fragment Per Kilobase of exon per Million fragments mapped (FPKM). The downloaded UCSC GTF file was passed to Cuffdiff along with the original alignment (SAM) file produced by Tophat. Cuffdiff re-estimates the abundance of the transcripts listed in the GTF file using alignments from the SAM file and concurrently test for different expression. The |log2(flod change|≥1 and p value < 0.05 was set as the threshold for significantly differential expression. The information of basic sequencing data is listed in supplementary materials: S1 Data.
- Martin, M. Cut adapt removes adapter sequences from high-throughput sequencing reads. Embnet Journal 2011, 17, 10-12, doi:10.14806/ej.17.1.200.
- Trapnell, C.; Pachter, L.; Salzberg, S.L. TopHat: discovering splice junctions with RNA-Seq. Bioinformatics 2009, 25, 1105-1111, doi:10.1093/bioinformatics/btp120.
- Trapnell, C.; Williams, B.A.; Pertea, G.; Mortazavi, A.; Kwan, G.; van Baren, M.J.; Salzberg, S.L.; Wold, B.J.; Pachter, L. Transcript assembly and quantification by RNA-Seq reveals unannotated transcripts and isoform switching during cell differentiation. Nat Biotechnol 2010, 28, 511-515, doi:10.1038/nbt.1621.
- Kong, L.; Zhang, Y.; Ye, Z.Q.; Liu, X.Q.; Zhao, S.Q.; Wei, L.; Gao, G. CPC: assess the protein-coding potential of transcripts using sequence features and support vector machine. Nucleic Acids Res 2007, 35, W345-349, doi:10.1093/nar/gkm391.
- Sun, L.; Luo, H.; Bu, D.; Zhao, G.; Yu, K.; Zhang, C.; Liu, Y.; Chen, R.; Zhao, Y. Utilizing sequence intrinsic composition to classify protein-coding and long non-coding transcripts. Nucleic Acids Res 2013, 41, e166, doi:10.1093/nar/gkt646.
LncRNA
LC-MS Analysis
The peptide samples were diluted to 1 μg/μL on the machine buffet. Set the sample volume to 5μL and collect the scan mode for 60 minutes. Scan the peptides with a mass-to-charge ratio of 350-1200 in the sample. The mass spectrometry data was collected using the Triple TOF 5600 + LC/MS system (AB SCIEX, USA). The peptide samples were dissolved in 2% acetonitrile/0.1% formic acid, and analyzed using the Triple TOF 5600 plus mass spectrometer coupled with the Eksigent nano LC system (AB SCIEX, USA). The peptide solution was added to the C18 capture column (3 μm, 350 μm×0.5 mm, AB Sciex, USA), and the C18 analytical column (3 μm, 75μm×150) was applied with a 60 min time gradient and a flow rate of 300 nL/min. mm, Welch Materials, Inc) for gradient elution. The two mobile phases are buffer A (2% acetonitrile/0.1% formic acid/98% H2O) and buffer B (98% acetonitrile/0.1% formic acid/2% H2O). For IDA (Information Dependent Acquisition), the MS spectrum is scanned with anion accumulation time of 250 ms, and the MS spectrum of 30 precursor ions is acquired with anion accumulation time of 50 ms. Collect MS1 spectrum in the range of 350-1200 m/z, and collect MS2 spectrum in the range of 100-1500 m/z. Set the precursor ion dynamic exclusion time to 15 s.
Submit the original MS/MS files from the mass spectrometer to ProteinPilot (https://sciex.com.cn/products/software/proteinpilot-software, version 4.5, SCIEX, Redwood City, California, USA) for data analysis. For protein identification, the Paragon algorithm in ProteinPilot was used to search the Bovine_uniprot database. The parameters are set as follows: the instrument is TripleTOF 5600, and the cysteine is modified with iodoacetamide; the biological modification is selected as the ID focus. For the identified protein results, select certain filtering criteria, and peptides with an unused score> 1.3 (a credibility of more than 95%) are considered credible peptides, and proteins containing at least one unique peptide are retained. The information of basic protein data is listed in supplementary materials: S2 Data.
miRNA
LC-MS Analysis
LC-MS analysis was performed on a nano Acquity UPLC system (Waters Corporation, USA) connected to a LTQ Orbitrap XL mass spectrometer (Thermo Scientific, Bremen, Germany) equipped with an online nano-electrospray ion source (Michrom Bioresources, USA). Peptides were resuspended with 20 µl solvent A (5% acetonitrile, 0.1% formic acid in water). 18 µl peptide solution was loaded onto the trap column (100μm × 20mm Acclaim PepMap C18, Thermo Fisher Scientific, San Jose, CA, USA) at a 20 μl/min flow rate of solvent A for 3 min and then was separated on a Acclaim PepMap C18 reverse phase column (75μm ×150mm, Thermo Fisher Scientific, USA) with a linear gradient. The gradient started from 2% solvent B (90% acetonitrile, 0.1% formic acid in water) to 40% solvent B over 105 min. The column was re-equilibrated at initial conditions for 15 min. The column flow rate was maintained at 300 nL/min and column temperature was maintained at 40 ℃. The electrospray voltage of 1.9 kV versus the inlet of the mass spectrometer was used.
LTQ Orbitrap XL mass spectrometer was operated in the data-dependent mode to switch automatically between MS and MS/MS acquisition. Survey of full-scan MS spectra with one microscan (m/z 300-1800) was acquired in the Obitrap with a mass resolution of 60,000 at m/z 400, followed by MS/MS of the eight most-intense peptide ions in the LTQ analyzer. The automatic gain control (AGC) was set to 1000 000 ions, with maximum accumulation times of 500 ms. For MS/MS, we used an isolation window of 2 m/z and the automatic gain control (AGC) of LTQ was set to 20 000 ions, with maximum accumulation time of 120 ms. Single charge state was rejected and dynamic exclusion was used with two micro scans in 10 s and 90 s exclusion duration. For MS/MS, precursor ions were activated using 35% normalized collision energy at the default activation q of 0.25 and an activation time of 30 ms. The spectrum were recorded with Xcalibur (version 2.2.0) software.
The mass spectra generated by the LTQ-XL instrument were processed using MaxQuant software (version 1.4.0.8, http://www.maxquant.org/). Data were searched using the Andromeda search engine against Bovine_uniprot database. The parameters for the database search were set as follows: (1) The minimum required peptide length was seven amino acids. (2) Trypsin cleavage specificity was applied with up to one missed cleavage allowed. (3) Variable modifications included methionine oxidation (M). (4) The mass tolerance for precursor and fragment ions was set to 20 ppm and 0.6 Da, respectively. (5) The false discovery rate (FDR) was set to 1% at both the peptide and protein levels. (6) Only proteins sequenced with at least two peptides were considered a reliable identification. The information of basic protein data is listed in supplementary materials: S3 Data.
Thank you again for your hard work. We look forward to receiving your reply.